# Fast Mixing Markov Chains for Strongly Rayleigh Measures, DPPs, and Constrained Sampling

**Chengtao Li**
MIT
ctli@mit.edu

**Stefanie Jegelka**
MIT
stefje@csail.mit.edu

**Suvrit Sra**
MIT
suvrit@mit.edu

## Abstract

We study probability measures induced by set functions with constraints. Such measures arise in a variety of real-world settings, where prior knowledge, resource limitations, or other pragmatic considerations impose constraints. We consider the task of rapidly sampling from such constrained measures, and develop fast Markov chain samplers for them. Our first main result is for MCMC sampling from Strongly Rayleigh (SR) measures, for which we present sharp polynomial bounds on the mixing time. As a corollary, this result yields a fast mixing sampler for Determinantal Point Processes (DPPs), yielding (to our knowledge) the first provably fast MCMC sampler for DPPs since their inception over four decades ago. Beyond SR measures, we develop MCMC samplers for probabilistic models with hard constraints and identify sufficient conditions under which their chains mix rapidly. We illustrate our claims by empirically verifying the dependence of mixing times on the key factors governing our theoretical bounds.

## 1 Introduction

Distributions over subsets of objects arise in a variety of machine learning applications. They occur as discrete probabilistic models [5, 20, 28, 36, 38] in computer vision, computational biology and natural language processing. They also occur in combinatorial bandit learning [9], as well as in recent applications to neural network compression [32] and matrix approximations [29].

Yet, practical use of discrete distributions can be hampered by computational challenges due to their combinatorial nature. Consider for instance sampling, a task fundamental to learning, optimization, and approximation. Without further restrictions, efficient sampling can be impossible [13]. Several lines of work thus focus on identifying tractable sub-classes, which in turn have had wide-ranging impacts on modeling and algorithms. Important examples include the Ising model [22], matchings (and the matrix permanent) [23], spanning trees (and graph algorithms) [2, 6, 16, 37], and Determinantal Point Processes (DPPs) that have gained substantial attention in machine learning [3, 17, 24, 26, 28, 30].

In this work, we extend the classes of tractable discrete distributions. Specifically, we consider the following two classes of distributions on $2^V$ (the set of subsets of a ground set $V = [N] := \{1, \ldots, N\}$): (1) strongly Rayleigh (SR) measures, and (2) distributions with certain cardinality or matroid-constraints. We analyze Markov chains for sampling from both classes. As a byproduct of our analysis, we answer a long-standing question about rapid mixing of MCMC sampling from DPPs.

SR measures are defined by strong negative correlations, and have recently emerged as valuable tools in the design of algorithms [2], in the theory of polynomials and combinatorics [4], and in machine learning through DPPs, a special case of SR distributions. Our first main result is the first polynomial-time sampling algorithm that applies to all SR measures (and thus *a fortiori* to DPPs).

General distributions on $2^V$ with constrained support (case (2) above) typically arise upon incorporating prior knowledge or resource constraints. We focus on resource constraints such as bounds on

cardinality and bounds on including limited items from sub-groups. Such constraints can be phrased as a family $\mathcal{C} \subseteq 2^V$ of subsets; we say $S$ satisfies the constraint $\mathcal{C}$ iff $S \in \mathcal{C}$. Then the distribution of interest is of the form

$$\pi_{\mathcal{C}}(S) \propto \exp(\beta F(S))[\![S \in \mathcal{C}]\!], \tag{1.1}$$

where $F : 2^V \to \mathbf{R}$ is a set function that encodes relationships between items $i \in V$, $[\![\cdot]\!]$ is the Iverson bracket, and $\beta$ a constant (also referred to as the inverse *temperature*). Most prior work on sampling with combinatorial constraints (such as sampling the bases of a matroid), assumes that $F$ breaks up linearly using element-wise weights $w_i$, i.e., $F(S) = \sum_{i \in S} w_i$. In contrast, we allow generic, nonlinear functions, and obtain a mixing times governed by structural properties of $F$.

**Contributions.** We briefly summarize the key contributions of this paper below.

– We derive a provably fast mixing Markov chain for efficient sampling from strongly Rayleigh measure $\pi$ (Theorem 2). This Markov chain is novel and may be of independent interest. Our results provide the first polynomial guarantee (to our knowledge) for Markov chain sampling from a general DPP, and more generally from an SR distribution.[1]

– We analyze (Theorem 4) mixing times of an exchange chain when the constraint family $\mathcal{C}$ is the set of bases of a special matroid, i.e., $|S| = k$ or $S$ obeys a partition constraint. Both of these constraints have high practical relevance [25, 27, 38].

– We analyze (Theorem 6) mixing times of an add-delete chain for the case $|S| \leq k$, which, perhaps surprisingly, turns out to be quite different from $|S| = k$. This constraint can be more practical than the strict choice $|S| = k$, because in many applications, the user may have an upper bound on the budget, but may not necessarily want to expend all $k$ units.

Finally, a detailed set of experiments illustrates our theoretical results.

**Related work.** Recent work in machine learning addresses sampling from distributions with sub- or supermodular $F$ [19, 34], determinantal point processes [3, 29], and sampling by optimization [14, 31]. Many of these works (necessarily) make additional assumptions on $\pi_{\mathcal{C}}$, or are approximate, or cannot handle constraints. Moreover, the constraints cannot easily be included in $F$: an out-of-the-box application of the result in [19], for instance, would lead to an unbounded constant in the mixing time.

Apart from sampling, other related tracts include work on variational inference for combinatorial distributions [5, 11, 36, 38] and inference for submodular processes [21]. Special instances of (1.1) include [27], where the authors limit DPPs to sets that satisfy $|S| = k$; partition matroid constraints are studied in [25], while the budget constraint $|S| \leq k$ has been used recently in learning DPPs [17]. Important existing results show fast mixing for a sub-family of strongly Rayleigh distributions [3, 15]; but those results do not include, for instance, general DPPs.

## 1.1 Background and Formal Setup

Before describing the details of our new contributions, let us briefly recall some useful background that also serves to set the notation. Our focus is on sampling from $\pi_{\mathcal{C}}$ in (1.1); we denote by $Z = \sum_{S \subseteq V} \exp(\beta F(S))$ and $Z_{\mathcal{C}} = \sum_{S \subseteq \mathcal{C}} \exp(\beta F(S))$. The simplest example of $\pi_{\mathcal{C}}$ is the uniform distribution over sets in $\mathcal{C}$, where $F(S)$ is constant. In general, $F$ may be highly nonlinear.

We sample from $\pi_{\mathcal{C}}$ using MCMC, i.e., we run a Markov Chain with state space $\mathcal{C}$. All our chains are ergodic. The *mixing time* of the chain indicates the number of iterations $t$ that we must perform (after starting from an arbitrary set $X_0 \in \mathcal{C}$) before we can consider $X_t$ as a valid sample from $\pi_{\mathcal{C}}$. Formally, if $\delta_{X_0}(t)$ is the total variation distance between the distribution of $X_t$ and $\pi_{\mathcal{C}}$ after $t$ steps, then $\tau_{X_0}(\varepsilon) = \min\{t : \delta_{X_0}(t') \leq \varepsilon, \ \forall t' \geq t\}$ is the mixing time to sample from a distribution $\epsilon$-close to $\pi_{\mathcal{C}}$ in terms of total variation distance. We say that the chain mixes fast if $\tau_{X_0}$ is polynomial in $N$. The mixing time can be bounded in terms of the eigenvalues of the transition matrix, as the following classic result shows:

**Theorem 1** (Mixing Time [10]). *Let $\lambda_i$ be the eigenvalues of the transition matrix, and $\lambda_{\max} = \max\{\lambda_2, |\lambda_N|\} < 1$. Then, the mixing time starting from an initial set $X_0 \in \mathcal{C}$ is bounded as*

$$\tau_{X_0}(\varepsilon) \leq (1 - \lambda_{\max})^{-1}(\log \pi_{\mathcal{C}}(X_0)^{-1} + \log \varepsilon^{-1}).$$

Most of the effort in bounding mixing times hence is devoted to bounding this eigenvalue.

## 2   Sampling from Strongly Rayleigh Distributions

In this section, we consider sampling from *strongly Rayleigh (SR)* distributions. Such distributions capture the strongest form of negative dependence properties, while enjoying a host of other remarkable properties [4]. For instance, they include the widely used DPPs as a special case. A distribution is SR if its generating polynomial $p_\pi : \mathbb{C}^N \to \mathbb{C}$, $p_\pi(z) = \sum_{S \subseteq V} \pi(S) \prod_{i \in S} z_i$ is *real stable*. This means if $\Im(z_i) > 0$ for all arguments $z_i$ of $p_\pi(z)$, then $p_\pi(z) > 0$.

We show in particular that SR distributions are amenable to efficient Markov chain sampling. Our starting point is the observation of [4] on closure properties of SR measures; of these we use *symmetric homogenization*. Given a distribution $\pi$ on $2^{[N]}$, its symmetric homogenization $\pi_{sh}$ on $2^{[2N]}$ is

$$\pi_{sh}(S) := \begin{cases} \pi(S \cap [N]) \binom{N}{S \cap [N]}^{-1} & \text{if } |S| = N; \\ 0 & \text{otherwise.} \end{cases}$$

If $\pi$ is SR, so is $\pi_{sh}$. We use this property below in our derivation of a fast-mixing chain.

We use here a recent result of Anari et al. [3], who show a Markov chain that mixes rapidly for *homogeneous SR* distributions. These distributions are over all subsets $S \subseteq V$ of some fixed size $|S| = k$, and hence do not include general DPPs. Concretely, for any $k$-homogeneous SR distribution $\pi : \{0,1\}^N \to \mathbb{R}_+$, a Gibbs-exchange sampler has mixing time

$$\tau_{X_0}(\varepsilon) \leq 2k(N-k)(\log \pi(X_0)^{-1} + \log \varepsilon^{-1}).$$

This sampler uniformly samples one item in the current set, and one outside the current set, and swaps them with an appropriate probability. Using these ideas we show how to obtain fast mixing chains for *any* general SR distribution $\pi$ on $[N]$. First, we construct its symmetric homogenization $\pi_{sh}$, and sample from $\pi_{sh}$ using a Gibbs-exchange sampler. This chain is fast mixing, thus we will efficiently get a sample $T \sim \pi_{sh}$. The corresponding sample for $\pi$ can be then obtained by computing $S = T \cap V$. Theorem 2, proved in the appendix, formally establishes the validity of this idea.

**Theorem 2.** *If $\pi$ is SR, then the mixing time of a Gibbs-exchange sampler for $\pi_{sh}$ is bounded as*

$$\tau_{X_0}(\varepsilon) \leq 2N^2 \Big( \log \binom{N}{|X_0|} + \log(\pi(X_0))^{-1} + \log \varepsilon^{-1} \Big). \tag{2.1}$$

For Theorem 2 we may choose the initial set such that $X_0$ makes the first term in the sum logarithmic in $N$ ($X_0 = T_0 \cap V$ in Algorithm 1).

---

**Algorithm 1** Markov Chain for Strongly Rayleigh Distributions

---

**Require:** SR distribution $\pi$
  Initialize $T \subseteq [2N]$ where $|T| = N$ and take $S = T \cap V$
  **while** not mixed **do**
    Draw $q \sim \text{Unif}[0,1]$
    Draw $t \in V \backslash S$ and $s \in S$ uniformly at random
    **if** $q \in [0, \frac{(N-|S|)^2}{2N^2})$ **then**
      $S = S \cup \{t\}$ with probability $\min\{1, \frac{\pi(S \cup \{t\})}{\pi(S)} \times \frac{|S|+1}{N-|S|}\}$                 ▷ Add $t$
    **else if** $q \in [\frac{(N-|S|)^2}{2N^2}, \frac{N-|S|}{2N})$ **then**
      $S = S \cup \{t\} \backslash \{s\}$ with probability $\min\{1, \frac{\pi(S \cup \{t\} \backslash \{s\})}{\pi(S)}\}$         ▷ Exchange $s$ with $t$
    **else if** $q \in [\frac{N-|S|}{2N}, \frac{|S|^2 + N(N-|S|)}{2N^2})$ **then**
      $S = S \backslash \{s\}$ with probability $\min\{1, \frac{\pi(S \backslash \{s\})}{\pi(S)} \times \frac{|S|}{N-|S|+1}\}$            ▷ Delete $s$
    **else**
      Do nothing
    **end if**
  **end while**

---

**Efficient Implementation.** Directly running a chain to sample $N$ items from a (doubled) set of size $2N$ adds some computational overhead. Hence, we construct an equivalent, more space-efficient

chain (Algorithm 1) on the initial ground set $V = [N]$ that only maintains $S \subseteq V$. Interestingly, this sampler is a mixture of add-delete and Gibbs-exchange samplers. This combination makes sense intuitively, too: add-delete moves (also shown in Alg. 3) are needed since the exchange sampler cannot change the cardinality of $S$. But a pure add-delete chain can stall if the sets concentrate around a fixed cardinality (low probability of a larger or smaller set). Exchange moves will not suffer the same high rejection rates. The key idea underlying Algorithm 1 is that the elements in $\{N + 1, \ldots, 2N\}$ are indistinguishable, so it suffices to maintain merely the cardinality of the currently selected subset instead of all its indices. Appendix C contains a detailed proof.

**Corollary 3.** *The bound* (2.1) *applies to the mixing time of Algorithm 1.*

**Remarks.** By assuming $\pi$ is SR, we obtain a clean bound for fast mixing. Compared to the bound in [19], our result avoids the somewhat opaque factor $\exp(\beta \zeta_F)$ that depends on $F$.

In certain cases, the above chain may mix slower in practice than a pure add-delete chain that was used in previous works [19, 24], since its probability of doing nothing is higher. In other cases, it mixes much faster than the pure add-delete chain; we observe both phenomena in our experiments in Sec. 4. Contrary to a simple add-delete chain, in all cases, it is *guaranteed* to mix well.

## 3 Sampling from Matroid-Constrained Distributions

In this section we consider sampling from an explicitly-constrained distribution $\pi_{\mathcal{C}}$ where $\mathcal{C}$ specifies certain matroid base constraints (§3.1) or a uniform matroid of a given rank (§3.2).

### 3.1 Matroid Base Constraints

We begin with constraints that are special cases of matroid bases[2]:

1. *Uniform matroid:* $\mathcal{C} = \{S \subseteq V \mid |S| = k\}$,
2. *Partition matroid:* Given a partition $V = \bigcup_{i=1}^{k} \mathcal{P}_i$, we allow sets that contain exactly one element from each $\mathcal{P}_i$: $\mathcal{C} = \{S \subseteq V \mid |S \cap \mathcal{P}_i| = 1 \text{ for all } 1 \leq i \leq k\}$.

An important special case of a distribution with a uniform matroid constraint is the $k$-DPP [27]. Partition matroids are used in multilabel problems [38], and also in probabilistic diversity models [21].

---

**Algorithm 2** Gibbs Exchange Sampler for Matroid Bases

---

**Require:** set function $F$, $\beta$, matroid $\mathcal{C} \subseteq 2^V$
  Initialize $S \in \mathcal{C}$
  **while** not mixed **do**
    Let $b = 1$ with probability 0.5
    **if** $b = 1$ **then**
      Draw $s \in S$ and $t \in V \backslash S$ ($t \in \mathcal{P}(s) \backslash \{s\}$) uniformly at random
      **if** $S \cup \{t\} \backslash \{s\} \in \mathcal{C}$ **then**
        $S \leftarrow S \cup \{t\} \backslash \{s\}$ with probability $\frac{\pi_{\mathcal{C}}(S \cup \{t\} \backslash \{s\})}{\pi_{\mathcal{C}}(S) + \pi_{\mathcal{C}}(S \cup \{t\} \backslash \{s\})}$
      **end if**
    **end if**
  **end while**

---

The sampler is shown in Algorithm 2. At each iteration, we randomly select an item $s \in S$ and $t \in V \backslash S$ such that the new set $S \cup \{t\} \backslash \{s\}$ satisfies $\mathcal{C}$, and swap them with certain probability. For uniform matroids, this means $t \in V \backslash S$; for partition matroids, $t \in \mathcal{P}(s) \backslash \{s\}$ where $\mathcal{P}(s)$ is the part that $s$ resides in. The fact that the chain has stationary distribution $\pi_{\mathcal{C}}$ can be inferred via detailed balance. Similar to the analysis in [19] for *unconstrained* sampling, the mixing time depends on a quantity that measures how much $F$ deviates from linearity: $\zeta_F = \max_{S,T \in \mathcal{C}} |F(S) + F(T) - F(S \cap T) - F(S \cup T)|$. Our proof, however, differs from that of [19]. While they use canonical paths [10], we use multicommodity flows, which are more effective in our constrained setting.

**Theorem 4.** *Consider the chain in Algorithm 2. For the uniform matroid, $\tau_{X_0}(\varepsilon)$ is bounded as*

$$\tau_{X_0}(\varepsilon) \leq 4k(N - k) \exp(\beta(2\zeta_F))(\log \pi_{\mathcal{C}}(X_0)^{-1} + \log \varepsilon^{-1}); \qquad (3.1)$$

*For the partition matroid, the mixing time is bounded as*

$$\tau_{X_0}(\varepsilon) \leq 4k^2 \max_i |\mathcal{P}_i| \exp(\beta(2\zeta_F))(\log \pi_{\mathcal{C}}(X_0)^{-1} + \log \varepsilon^{-1}). \tag{3.2}$$

Observe that if $\mathcal{P}_i$'s form an equipartition, i.e., $|\mathcal{P}_i| = N/k$ for all $i$, then the second bound becomes $\widetilde{\mathcal{O}}(kN)$. For $k = \mathcal{O}(\log N)$, the mixing times depend as $\mathcal{O}(N\mathrm{polylog}(N)) = \widetilde{\mathcal{O}}(N)$ on $N$. For uniform matroids, the time is equally small if $k$ is close to $N$. Finally, the time depends on the initialization, $\pi_{\mathcal{C}}(X_0)$. If $F$ is monotone increasing, one may run a simple greedy algorithm to ensure that $\pi_{\mathcal{C}}(X_0)$ is large. If $F$ is monotone submodular, this ensures that $\log \pi_{\mathcal{C}}(X_0)^{-1} = \mathcal{O}(\log N)$.

Our proof uses a multicommodity flow to upper bound the largest eigenvalue of the transition matrix. Concretely, let $\mathcal{H}$ be the set of all simple paths between states in the state graph of Markov chain, we construct a flow $f : \mathcal{H} \to \mathbb{R}^+$ that assigns a nonnegative flow value to any simple path between any two states (sets) $X, Y \in \mathcal{C}$. Each edge $e = (S, T)$ in the graph has a capacity $Q(e) = \pi_{\mathcal{C}}(S)P(S, T)$ where $P(S, T)$ is the transition probability from $S$ to $T$. The total flow sent from $X$ to $Y$ must be $\pi_{\mathcal{C}}(X)\pi_{\mathcal{C}}(Y)$: if $\mathcal{H}_{XY}$ is the set of all simple paths from $X$ to $Y$, then we need $\sum_{p \in \mathcal{H}_{XY}} f(p) = \pi_{\mathcal{C}}(X)\pi_{\mathcal{C}}(Y)$. Intuitively, the mixing time relates to the congestion in any edge, and the length of the paths. If there are many short paths $X \rightsquigarrow Y$ across which flow can be distributed, then mixing is fast. This intuition is captured in a fundamental theorem:

**Theorem 5** (Multicommodity Flow [35]). *Let $E$ be the set of edges in the transition graph, and $P(X, Y)$ the transition probability. Define*

$$\overline{\rho}(f) = \max_{e \in E} \frac{1}{Q(e)} \sum_{p \ni e} f(p)\mathrm{len}(p),$$

*where* $\mathrm{len}(p)$ *the length of the path $p$. Then $\lambda_{\max} \leq 1 - 1/\overline{\rho}(f)$.*

With this property of multicommodity flow, we are ready to prove Thm. 4.

*Proof. (Theorem 4)* We sketch the proof for partition matroids; the full proofs is in Appendix A. For any two sets $X, Y \in \mathcal{C}$, we distribute the flow equally across all shortest paths $X \rightsquigarrow Y$ in the transition graph and bound the amount of flow through any edge $e \in E$.

Consider two arbitrary sets $X, Y \in \mathcal{C}$ with symmetric difference $|X \oplus Y| = 2m \leq 2k$, i.e., $m$ elements need to be exchanged to reach from $X$ to $Y$. However, these $m$ steps are a valid path in the transition graph only if every set $S$ along the way is in $\mathcal{C}$. The exchange property of matroids implies that this requirement is indeed true, so any shortest path $X \rightsquigarrow Y$ has length $m$. Moreover, there are exactly $m!$ such paths, since we can exchange the elements in $X \setminus Y$ in any order to reach at $Y$. Note that once we choose $s \in X \setminus Y$ to swap out, there is only one choice $t \in Y \setminus X$ to swap in, where $t$ lies in the same part as $s$ in the partition matroid, otherwise the constraint will be violated. Since the total flow is $\pi_{\mathcal{C}}(X)\pi_{\mathcal{C}}(Y)$, each path receives $\pi_{\mathcal{C}}(X)\pi_{\mathcal{C}}(Y)/m!$ flow.

Next, let $e = (S, T)$ be any edge on some shortest path $X \rightsquigarrow Y$; so $S, T \in \mathcal{C}$ and $T = S \cup \{j\} \setminus \{i\}$ for some $i, j \in V$. Let $2r = |X \oplus S| < 2m$ be the length of the shortest path $X \rightsquigarrow S$, i.e., $r$ elements need to be exchanged to reach from $X$ to $S$. Similarly, $m - r - 1$ elements are exchanged to reach from $T$ to $Y$. Since there is a path for every permutation of those elements, the ratio of the total flow $w_e(X, Y)$ that edge $e$ receives from pair $X, Y$, and $Q(e)$, becomes

$$\frac{w_e(X, Y)}{Q(e)} \leq \frac{2r!(m-1-r)!kL}{m!Z_{\mathcal{C}}} \exp(2\beta\zeta_F)(\exp(\beta F(\sigma_S(X, Y))) + \exp(\beta F(\sigma_T(X, Y)))), \tag{3.3}$$

where we define $\sigma_S(X, Y) = X \oplus Y \oplus S = (X \cap Y \cap S) \cup (X \setminus (Y \cup S)) \cup (Y \setminus (X \cup S))$. To bound the total flow, we must count the pairs $X, Y$ such that $e$ is on their shortest path(s), and bound the flow they send. We do this in two steps, first summing over all $(X, Y)$'s that share the upper bound (3.3) since they have the same difference sets $U_S = \sigma_S(X, Y)$ and $U_T = \sigma_T(X, Y)$, and then we sum over all possible $U_S$ and $U_T$. For fixed $U_S, U_T$, there are $\binom{m-1}{r}$ pairs that share those difference sets, since the only freedom we have is to assign $r$ of the $m - 1$ elements in $S \setminus (X \cap Y \cap S)$ to $Y$, and the rest to $X$. Hence, for fixed $U_S, U_T$. Appropriate summing and canceling then yields

$$\sum_{\substack{(X,Y): \sigma_S(X,Y)=U_S, \\ \sigma_T(X,Y)=U_T}} \frac{w_e(X, Y)}{Q(e)} \leq \frac{2kL}{Z_{\mathcal{C}}} \exp(2\beta\zeta_F)(\exp(\beta F(U_S)) + \exp(\beta F(U_T))). \tag{3.4}$$

Finally, we sum over all valid $U_S$ ($U_T$ is determined by $U_S$). One can show that any valid $U_S \in \mathcal{C}$, and hence $\sum_{U_S} \exp(\beta F(U_S)) \leq Z_\mathcal{C}$, and likewise for $U_T$. Hence, summing the bound (3.4) over all possible choices of $U_S$ yields

$$\bar{\rho}(f) \leq 4kL \exp(2\beta \zeta_F) \max_p \mathrm{len}(p) \leq 4k^2 L \exp(2\beta \zeta_F),$$

where we upper bound the length of any shortest path by $k$, since $m \leq k$. Hence

$$\tau_{X_0}(\varepsilon) \leq 4k^2 L \exp(2\beta \zeta_F)(\log \pi(X_0)^{-1} + \log \varepsilon^{-1}). \qquad \square$$

For more restrictive constraints, there are fewer paths, and the bounds can become larger. Appendix A shows the general dependence on $k$ (as $k!$). It is also interesting to compare the bound on uniform matroid in Eq. (3.1) to that shown in [3] for a sub-class of distributions that satisfy the property of being homogeneous strongly Rayleigh[3]. If $\pi_\mathcal{C}$ is homogeneous strongly Rayleigh, we have $\tau_{X_0}(\varepsilon) \leq 2k(N-k)(\log \pi_\mathcal{C}(X_0)^{-1} + \log \varepsilon^{-1})$. In our analysis, without additional assumptions on $\pi_\mathcal{C}$, we pay a factor of $2 \exp(2\beta \zeta_F))$ for generality. This factor is one for some strongly Rayleigh distributions (e.g., if $F$ is modular), but not for all.

### 3.2 Uniform Matroid Constraint

We consider constraints that is a uniform matroid of certain rank: $\mathcal{C} = \{S : |S| \leq k\}$. We employ the lazy add-delete Markov chain in Algo. 3, where in each iteration, with probability 0.5 we uniformly randomly sample one element from $V$ and either add it to or delete it from the current set, while respecting constraints. To show fast mixing, we consider using *path coupling*, which essentially says that if we have a contraction of two (coupling) chains then we have fast mixing. We construct path coupling $(S, T) \to (S', T')$ on a carefully generated graph with edges $E$ (from a proper metric). With all details in Appendix B we end up with the following theorem:

**Theorem 6.** *Consider the chain shown in Algorithm 3. Let $\alpha = \max_{(S,T) \in E} \{\alpha_1, \alpha_2\}$ where $\alpha_1$ and $\alpha_2$ are functions of edges $(S, T) \in E$ and are defined as*

$$\alpha_1 = 1 - \sum_{i \in T} |p^-(T, i) - p^-(S, i)|_+ - [\![|S| < k]\!] \sum_{i \in [N] \setminus S} (p^+(S, i) - p^+(T, i))_+;$$

$$\alpha_2 = \min\{p^-(S, s), p^-(T, t)\} - \sum_{i \in R} |p^-(S, i) - p^-(T, i)| +$$

$$[\![|S| < k]\!](\min\{p^+(S, t), p^+(T, s)\} - \sum_{i \in [N] \setminus (S \cup T)} |p^+(S, i) - p^+(T, i)|),$$

*where $(x)_+ = \max(0, x)$. The summations over absolute differences quantify the sensitivity of transition probabilities to adding/deleting elements in neighboring $(S, T)$. Assuming $\alpha < 1$, we get*

$$\tau(\varepsilon) \leq \frac{2N \log(N \varepsilon^{-1})}{1 - \alpha}$$

---

**Algorithm 3** Gibbs Add-Delete Markov Chain for Uniform Matroid

---

**Require:** $F$ the set function, $\beta$ the inverse temperature, $V$ the ground set, $k$ the rank of $\mathcal{C}$
**Ensure:** $S$ sampled from $\pi_\mathcal{C}$
  Initialize $S \in \mathcal{C}$
  **while** not mixed **do**
    Let $b = 1$ with probability 0.5
    **if** $b = 1$ **then**
      Draw $s \in V$ uniformly randomly
      **if** $s \notin S$ and $|S \cup \{s\}| \leq k$ **then**
        $S \leftarrow S \cup \{s\}$ with probability $p^+(S, s) = \frac{\pi_\mathcal{C}(S \cup \{s\})}{\pi_\mathcal{C}(S) + \pi_\mathcal{C}(S \cup \{s\})}$
      **else**
        $S \leftarrow S \setminus \{s\}$ with probability $p^-(S, s) = \frac{\pi_\mathcal{C}(S \setminus \{s\})}{\pi_\mathcal{C}(S) + \pi_\mathcal{C}(S \setminus \{s\})}$
      **end if**
    **end if**
  **end while**

---

**Remarks.** If $\alpha$ is less than 1 and independent of $N$, then the mixing time is nearly linear in $N$. The condition is conceptually similar to those in [29, 34]. The fast mixing requires both $\alpha_1$ and $\alpha_2$, specifically, the change in probability when adding or deleting single element to neighboring subsets, to be small. Such notion is closely related to the *curvature* of discrete set functions.

[3]Appendix C contains details about strongly Rayleigh distributions.

# 4   Experiments

We next empirically study the dependence of sampling times on key factors that govern our theoretical bounds. In particular, we run Markov chains on chain-structured Ising models on a partition matroid base and DPPs on a uniform matroid, and consider estimating marginal and conditional probabilities of a single variable. To monitor the convergence of Markov chains, we use *potential scale reduction factor* (PSRF) [7, 18] that runs several chains in parallel and compares within-chain variances to between-chain variances. Typically, PSRF is greater than 1 and will converge to 1 in the limit; if it is close to 1 we empirically conclude that chains have mixed well. Throughout experiments we run 10 chains in parallel for estimations, and declare "convergence" at a PSRF of 1.05.

We first focus on small synthetic examples where we can compute exact marginal and conditional probabilities. We construct a 20-variable chain-structured Ising model as

$$\pi_{\mathcal{C}}(S) \propto \exp\left(\beta\left(\left(\delta \sum_{i=1}^{19} w_i(s_i \oplus s_{i+1})\right) + (1-\delta)|S|\right)\right) \llbracket S \in \mathcal{C} \rrbracket,$$

where the $s_i$ are 0-1 encodings of $S$, and the $w_i$ are drawn uniformly randomly from $[0, 1]$. The parameters $(\beta, \delta)$ govern bounds on the mixing time via $\exp(2\beta\zeta_F)$; the smaller $\delta$, the smaller $\zeta_F$. $\mathcal{C}$ is a partition matroid of rank 5. We estimate conditional probabilities of one random variable conditioned on 0, 1 and 2 other variables and compare against the ground truth. We set $(\beta, \delta)$ to be $(1, 1)$, $(3, 1)$ and $(3, 0.5)$ and results are shown in Fig. 1. All marginals and conditionals converge to their true values, but with different speed. Comparing Fig. 1a against 1b, we observe that with fixed $\delta$, increase in $\beta$ slows down the convergence, as expected. Comparing Fig. 1b against 1c, we observe that with fixed $\beta$, decrease in $\delta$ speeds up the convergence, also as expected given our theoretical results. Appendix D.1 and D.2 illustrate the convergence of estimations under other $(\beta, \delta)$ settings.

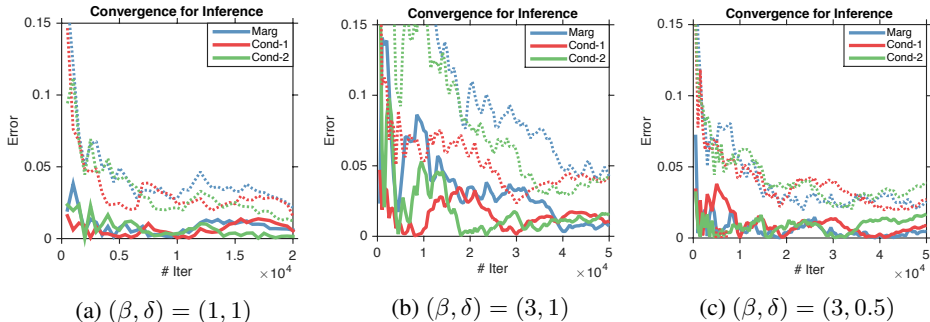

(a) $(\beta, \delta) = (1, 1)$　　　(b) $(\beta, \delta) = (3, 1)$　　　(c) $(\beta, \delta) = (3, 0.5)$

Figure 1: Convergence of marginal (`Marg`) and conditional (`Cond-1` and `Cond-2`, conditioned on 1 and 2 other variables) probabilities of a single variable in a 20-variable Ising model with different $(\beta, \delta)$. Full lines show the means and dotted lines the standard deviations of estimations.

We also check convergence on larger models. We use a DPP on a uniform matroid of rank 30 on the Ailerons data (`http://www.dcc.fc.up.pt/657~ltorgo/Regression/DataSets.html`) of size 200. Here, we do not have access to the ground truth, and hence plot the estimation mean with standard deviations among 10 chains in 3a. We observe that the chains will eventually converge, i.e., the mean becomes stable and variance small. We also use PSRF to approximately judge the convergence. More results can be found in Appendix D.3.

Furthermore, the mixing time depends on the size $N$ of the ground set. We use a DPP on Ailerons and vary $N$ from 50 to 1000. Fig. 2a shows the PSRF from 10 chains for each setting. By thresholding PSRF at 1.05 in Fig. 2b we see a clearer dependence on $N$. At this scale, the mixing time grows almost linearly with $N$, indicating that this chain is efficient at least at small to medium scale.
Finally, we empirically study how fast our sampler on strongly Rayleigh distribution converges. We compare the chain in Algorithm 1 (`Mix`) against a simple add-delete chain (`Add-Delete`). We use a DPP on Ailerons data[4] of size 200, and the corresponding PSRF is shown in Fig. 3b. We observe that `Mix` converges slightly slower than `Add-Delete` since it is lazier. However, the Add-Delete chain does not always mix fast. Fig. 3c illustrates a different setting, where we modify the eigenspectrum of the kernel matrix: the first 100 eigenvalues are 500 and others 1/500. Such a kernel corresponds to

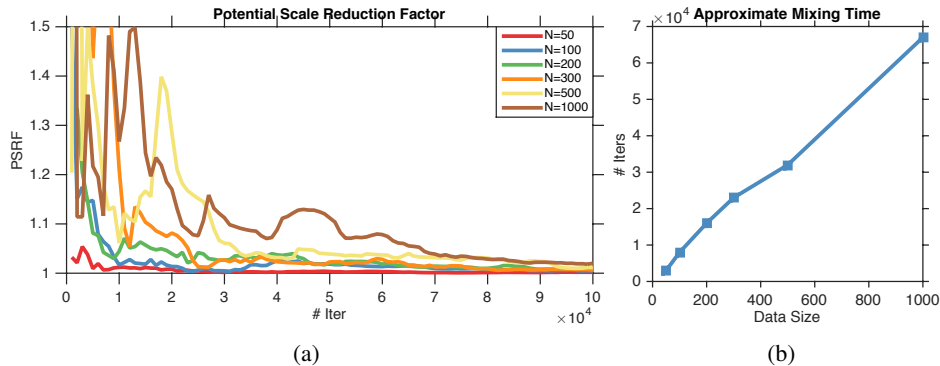

(a)                                         (b)

Figure 2: Empirical mixing time analysis when varying dataset sizes, (a) PSRF's for each set of chains, (b) Approximate mixing time obtained by thresholding PSRF at 1.05.

almost an elementary DPP, where the size of the observed subsets sharply concentrates around 100. Here, `Add-Delete` moves very slowly. `Mix`, in contrast, has the ability of exchanging elements and thus converges way faster than `Add-Delete`.

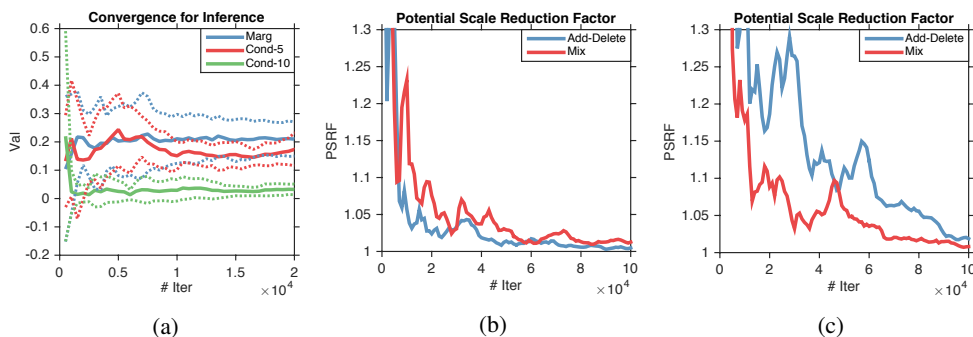

(a)                                         (b)                                         (c)

Figure 3: (a) Convergence of marginal and conditional probabilities by DPP on uniform matroid, (b,c) comparison between add-delete chain (Algorithm 3) and projection chain (Algorithm 1) for two instances: slowly decaying spectrum and sharp step in the spectrum.

## 5   Discussion and Open Problems

We presented theoretical results on Markov chain sampling for discrete probabilistic models subject to implicit and explicit constraints. In particular, under an implicit constraint that the probability measure is strongly Rayleigh, we obtain an unconditional fast mixing guarantee. For distributions with various explicit constraints we showed sufficient conditions for fast mixing. We show empirically that the dependencies of mixing times on various factors are consistent with our theoretical analysis.

There still exist many open problems in both implicitly- and explicitly-constrained settings. Many bounds that we show depend on structural quantities ($\zeta_F$ or $\alpha$) that may not always be easy to quantify in practice. It will be valuable to develop chains on special classes of distributions (like we did for strongly Rayleigh) whose mixing time is independent of these factors. Moreover, we only considered matroid bases or uniform matroids, while several important settings such as knapsack constraints remain open. In fact, even uniform sampling with a knapsack constraint is not easy; a mixing time of $\mathcal{O}(N^{4.5})$ is known [33]. We defer the development of similar or better bounds, potentially with structural factors like $\exp(\beta\zeta_F)$, on specialized discrete probabilistic models as our future work.

**Acknowledgements.** This research was partially supported by NSF CAREER 1553284 and a Google Research Award.

## Footnotes

[1]The analysis in [24] is not correct since it relies on a wrong construction of path coupling.

[2]Drawing even a uniform sample from the bases of an arbitrary matroid can be hard.

[4] `http://www.dcc.fc.up.pt/657~ltorgo/Regression/DataSets.html`

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
