[Supplementary Material · supp.pdf]

# A Proof of Thm. 4

## A.1 Proof for Uniform Matroid Base

*Proof.* We consider the case where $\mathcal{C}$ is uniform matroid base. For any two sets $X, Y \in \mathcal{C}$, we distribute the flow equally across all shortest paths $X \rightsquigarrow Y$ in the transition graph. Then, for arbitrary edge $e \in E$, we bound the number of paths (and flow) through $e$.

Consider two arbitrary sets $X, Y \in \mathcal{C}$ with symmetric difference $|X \oplus Y| = 2m \leq 2k$. Any shortest path $X \rightsquigarrow Y$ has length $m$. Moreover, there are exactly $(m!)^2$ such paths, since we can exchange the elements in $X \setminus Y$ in any order with the elements in $Y \setminus X$ in any order to reach at $Y$. Since the total flow is $\pi_{\mathcal{C}}(X)\pi_{\mathcal{C}}(Y)$, each path receives $\pi_{\mathcal{C}}(X)\pi_{\mathcal{C}}(Y)/(m!)^2$ flow.

Next, let $e = (S, T)$ be any edge on some shortest path $X \rightsquigarrow Y$; so $S, T \in \mathcal{C}$ and $T = S \cup \{j\}\setminus\{i\}$ for some $i, j \in [N]$. Let $2r = |X \oplus S| < 2m$ be the length of the shortest path $X \rightsquigarrow S$, thus there are $(r!)^2$ ways to reach from $X$ to $S$. Similarly, $m - r - 1$ elements are exchanged to reach from $T$ to $Y$ and there are in total $((m - r - 1)!)^2$ ways to do so. the total flow $e$ receives from pair $X, Y$ is

$$w_e(X, Y) = \frac{\pi_{\mathcal{C}}(X)\pi_{\mathcal{C}}(Y)}{(m!)^2}(r!)^2((m - 1 - r)!)^2$$

Since in our chain,

$$Q(e) = \frac{2Z_{\mathcal{C}}\exp(\beta F(S))\exp(\beta F(T))}{k(N - k)(\exp(\beta F(S)) + \exp(\beta F(T)))},$$

it follows that

$$\frac{w_e(X, Y)}{Q(e)} = \frac{2(r!)^2((m - 1 - r)!)^2 k(N - k)\exp(\beta(F(X) + F(Y)))(\exp(\beta F(S)) + \exp(\beta F(T)))}{(m!)^2 Z_{\mathcal{C}}\exp(\beta(F(S) + F(T)))}$$

$$\leq \frac{2(r!)^2((m - 1 - r)!)^2 k(N - k)}{(m!)^2 Z_{\mathcal{C}}}\exp(2\beta\zeta_F)(\exp(\beta F(\sigma_S(X, Y))) + \exp(\beta F(\sigma_T(X, Y)))),$$

where we define $\sigma_S(X, Y) = X \oplus Y \oplus S$. The inequality draws from the fact that

$$\frac{\exp(\beta(F(X) + F(Y) + F(S)))}{\exp(\beta(F(S) + F(T)))} = \exp(\beta(F(X) + F(Y) - F(T))$$

$$= \exp(\beta(F(X) + F(Y) - F(X \cap Y) - F(X \cup Y)))$$

$$\exp(\beta(F(X \cap Y) + F(X \cup Y) - F(T) - F(\sigma_T(X, Y))))\exp(\beta F(\sigma_T(X, Y)))$$

$$\leq \exp(2\beta\zeta_F)\exp(\beta F(\sigma_T(X, Y)))$$

and likewise for $\frac{\exp(\beta(F(X)+F(Y)+F(T)))}{\exp(\beta(F(S)+F(T)))}$. Similar trick has been used in [19].

Let $U_S = \sigma_S(X, Y)$ and $U_T = \sigma_T(X, Y)$, then for fixed $U_S, U_T$, the total flow that passes $e$ is

$$\sum_{\substack{(X,Y): \sigma_S(X,Y)=U_S, \\ \sigma_T(X,Y)=U_T}} \frac{w_e(X, Y)}{Q(e)}$$

$$\leq 2\sum_{r=0}^{m-1}\binom{m - 1}{r}^2 \frac{(r!)^2((m - 1 - r)!)^2 k(N - k)}{(m!)^2 Z}$$

$$\times \exp(2\beta\zeta_F)(\exp(\beta F(U_S)) + \exp(\beta F(U_T)))$$

$$= \frac{2k(N - k)}{mZ_{\mathcal{C}}}\exp(2\beta\zeta_F)(\exp(\beta F(U_S)) + \exp(\beta F(U_T))).$$

Finally, with the definition of $\overline{\rho}(f)$ we sum over all images of $U_S$ and $U_T$. Recall that $Z = \sum_{U_S}\exp(\beta F(U_S))$. Since $|S \oplus X \oplus Y| = k$ we know that $U_S, U_T \in \mathcal{C}$, thus $Z \leq Z_{\mathcal{C}}$ and

$$\overline{\rho}(f) \leq 4k(N - k)\exp(2\beta\zeta_F).$$

Hence

$$\tau_{X_0}(\varepsilon) \leq 4k(N - k)\exp(2\beta\zeta_F)(\log\pi_{\mathcal{C}}(X_0)^{-1} + \log\varepsilon^{-1}).$$

## A.2 Proof on Partition Matroid Base

*Proof.* Consider two arbitrary sets $X, Y \in \mathcal{C}$ with symmetric difference $|X \oplus Y| = 2m \leq 2k$, i.e., $m$ elements need to be exchanged to reach from $X$ to $Y$. However, these $m$ steps are a valid path in the transition graph only if every set $S$ along the way is in $\mathcal{C}$. The exchange property of matroids implies that this is indeed true, so any shortest path $X \rightsquigarrow Y$ has length $m$. Moreover, there are exactly $m!$ such paths, since we can exchange the elements in $X \setminus Y$ in any order to reach at $Y$. Note that once we choose $s \in X \setminus Y$ to swap out, there is only one choice $t \in Y \setminus X$ to swap in, where $t$ lies in the same part as $s$ in the partition matroid, otherwise the constraint will be violated. Since the total flow is $\pi_{\mathcal{C}}(X)\pi_{\mathcal{C}}(Y)$, each path receives $\pi_{\mathcal{C}}(X)\pi_{\mathcal{C}}(Y)/m!$ flow.

Next, let $e = (S, T)$ be any edge on some shortest path $X \rightsquigarrow Y$; so $S, T \in \mathcal{C}$ and $T = S \cup \{j\} \setminus \{i\}$ for some $i, j \in V$. Let $2r = |X \oplus S| < 2m$ be the length of the shortest path $X \rightsquigarrow S$, i.e., $r$ elements need to be exchanged to reach from $X$ to $S$. Similarly, $m - r - 1$ elements are exchanged to reach from $T$ to $Y$. Since there is a path for every permutation of those elements, the total flow edge $e$ receives from pair $X, Y$ is

$$w_e(X, Y) = \frac{\pi_{\mathcal{C}}(X)\pi_{\mathcal{C}}(Y)}{m!} r!(m - 1 - r)!.$$

Since, in our chain, (using $L = \max_i |\mathcal{P}_i| - 1$)

$$Q(e) \geq \frac{\pi_{\mathcal{C}}(S)}{2kL} \frac{\pi_{\mathcal{C}}(T)}{\pi_{\mathcal{C}}(S) + \pi_{\mathcal{C}}(T)} = \frac{\exp(\beta F(S))\exp(\beta F(T))}{2kLZ_{\mathcal{C}}(\exp(\beta F(S)) + \exp(\beta F(T)))},$$

it follows that

$$\frac{w_e(X, Y)}{Q(e)} \leq \frac{2r!(m - 1 - r)!kL\exp(\beta(F(X) + F(Y)))(\exp(\beta F(S)) + \exp(\beta F(T)))}{m!Z_{\mathcal{C}}\exp(\beta(F(S) + F(T)))}$$

$$\leq \frac{2r!(m - 1 - r)!kL}{m!Z_{\mathcal{C}}}\exp(2\beta\zeta_F)(\exp(\beta F(\sigma_S(X, Y))) + \exp(\beta F(\sigma_T(X, Y)))), \quad \text{(A.1)}$$

where we define $\sigma_S(X, Y) = X \oplus Y \oplus S = (X \cap Y \cap S) \cup (X \setminus (Y \cup S)) \cup (Y \setminus (X \cup S))$. To bound the total flow, we must count the pairs $X, Y$ such that $e$ is on their shortest path(s), and bound the flow they send. We do this in two steps, first summing over all $X, Y$ that share the upper bound (A.1) since they have the same difference sets $U_S = \sigma_S(X, Y)$ and $U_T = \sigma_T(X, Y)$, and then we sum over all possible $U_S$ and $U_T$. For fixed $U_S, U_T$, there are $\binom{m-1}{r}$ pairs that share those difference sets, since the only freedom we have is to assign $r$ of the $m - 1$ elements in $S \setminus (X \cap Y \cap S)$ to $Y$, and the rest to $X$. Hence, for fixed $U_S, U_T$:

$$\sum_{\substack{(X,Y):\, \sigma_S(X,Y)=U_S, \\ \sigma_T(X,Y)=U_T}} \frac{w_e(X, Y)}{Q(e)} \leq 2\sum_{r=0}^{m-1} \binom{m-1}{r} \frac{r!(m - 1 - r)!kL}{m!Z_{\mathcal{C}}}$$

$$\times \exp(2\beta\zeta_F)(\exp(\beta F(U_S)) + \exp(\beta F(U_T)))$$

$$= \frac{2kL}{Z_{\mathcal{C}}}\exp(2\beta\zeta_F)(\exp(\beta F(U_S)) + \exp(\beta F(U_T))). \quad \text{(A.2)}$$

Finally, we sum over all valid $U_S$ ($U_T$ is determined by $U_S$), where by "valid" we mean there exists $X, Y \in \mathcal{C}$ and $S \in \mathcal{C}$ on one path from $X$ to $Y$ such that, $U_S = \sigma_S(X, Y)$. Any such $U_S$ can be constructed by picking $k - m$ elements from $S$ (including $i$), and by replacing the remaining elements $u \in S$ by another member of their partition: i.e., if $u \in \mathcal{P}_\ell$, then it is replaced by some other $v \in \mathcal{P}_\ell$, since both $X$ and $Y$ must be in $\mathcal{C}$. Hence, any $U_S$ satisfies the partition constraint, i.e., $U_S \in \mathcal{C}$ and therefore $\sum_{U_S} \exp(\beta F(U_S)) \leq Z_{\mathcal{C}}$, and likewise for $U_T$. Hence, summing the bound (A.2) over all possible $U_S$ yields

$$\bar{\rho}(f) \leq 4kL\exp(2\beta\zeta_F)\max_p \text{len}(p) \leq 4k^2L\exp(2\beta\zeta_F),$$

where we upper bound the length of any shortest path by $k$, since $m \leq k$. Hence

$$\tau_{X_0}(\varepsilon) \leq 4k^2L\exp(2\beta\zeta_F)(\log \pi_{\mathcal{C}}(X_0)^{-1} + \log \varepsilon^{-1}). \qquad \square$$

## A.3 Proof for General Matroid Base

In the case where no structural assumption is made on $\mathcal{C}$, the proof needs to be more carefully handled. Because in this case, we know neither the number of legal paths between any two states, nor the number of $\sigma_S(X, Y)$ falls out of $\mathcal{C}$.

We again consider arbitrary sets $X, Y \in \mathcal{C}$ where $|X \oplus Y| = 2m \leq 2k$. The total number of shortest paths is *at least* $(m!)$ due to exchange property of matroids. Since the amount of flow from $X$ to $Y$ is $\pi_{\mathcal{C}}(X)\pi_{\mathcal{C}}(Y)$, each path receives *at most* $\pi_{\mathcal{C}}(x)\pi_{\mathcal{C}}(y)/m!$ .

Next, let $e = (S, T)$ be any edge on some shortest path $X \rightsquigarrow Y$; so $S, T \in \mathcal{C}$ and $T = S \cup \{j\}\backslash\{i\}$ for some $i, j \in V$. Let $2r = |X \oplus S| < 2m$ be the length of the shortest path $X \rightsquigarrow S$, thus there are at most $(r!)^2$ ways to reach from $X$ to $S$. Likewise there are at most $((m - r - 1)!)^2$ paths to reach from $T$ to $Y$. The total flow edge $e$ receives from pair $X, Y$ is then upper-bounded as

$$w_e(X, Y) \leq \frac{\pi_{\mathcal{C}}(X)\pi_{\mathcal{C}}(Y)}{m!}(r!)^2((m - 1 - r)!)^2.$$

It follows that

$$\frac{w_e(X, Y)}{Q(e)} \leq \frac{2(r!)^2((m - 1 - r)!)^2 k(N - k)}{m! Z_{\mathcal{C}}} \exp(2\beta\zeta_F)(\exp(\beta F(U_S)) + \exp(\beta F(U_T))).$$

The total pairs of $(X, Y)$ that passes $e$ with the same set of images is upper-bounded by $\binom{m-1}{r}^2$, thus the flow that passes $e$ with the same set of images is bounded as

$$\sum_{\substack{(X,Y): \sigma_S(X,Y)=U_S, \\ \sigma_T(X,Y)=U_T}} \frac{w_e(X, Y)}{Q(e)}$$

$$\leq 2 \sum_{r=0}^{m-1} \binom{m-1}{r}^2 \frac{(r!)^2((m - 1 - r)!)^2 k(N - k)}{m! Z}$$

$$\times \exp(2\beta\zeta_F)(\exp(\beta F(U_S)) + \exp(\beta F(U_T)))$$

$$= \frac{2(m-1)! k(N - k)}{Z_{\mathcal{C}}} \exp(2\beta\zeta_F)(\exp(\beta F(U_S)) + \exp(\beta F(U_T))).$$

Thus if we sum over all $U_S, U_T$, the result is upper-bounded as

$$\overline{\rho}(f) \leq \frac{4k! Z}{Z_{\mathcal{C}}} k(N - k) \exp(2\beta\zeta_F).$$

Note that here we upper-bounded $m$ with $k$ and $Z$ could be larger than $Z_{\mathcal{C}}$ because it may happen that $U_S \notin \mathcal{C}$. It follows that

$$\tau_{X_0}(\varepsilon) \leq \frac{4k! Z}{Z_{\mathcal{C}}} k(N - k) \exp(2\beta\zeta_F)(\log \pi_{\mathcal{C}}(X_0)^{-1} + \log \varepsilon^{-1}). \qquad \square$$

# B  Proof of Thm. 6

Assume we have a chain $(X_t)$ on state space $V$ with transition matrix $P$, a *coupling* is a new chain $(X_t, Y_t)$ on $V \times V$ such that both $(X_t)$ and $(Y_t)$, if considered marginally, are Markov chains with the same transition matrices $P$. The key point of coupling is to construct such a new chain to encourage $X_t$ and $Y_t$ to *coalesce* quickly. If, in the new chain, $\Pr(X_t \neq Y_t) \leq \varepsilon$ for some fixed $t$ regardless of the starting state $(X_0, Y_0)$, then $\tau(\varepsilon) \leq t$ [1]. To make the coupling construction easier, *Path coupling* [8] is then introduced so as to reduce the coupling to adjacent states in an appropriately constructed state graph. The coupling of arbitrary states follows by aggregation over a path between the two. Path coupling is formalized in the following lemma.

**Lemma 7.** *[8, 12] Let $\delta$ be an integer-valued metric on $V \times V$ where $\delta(\cdot, \cdot) \leq D$. Let $E$ be a subset of $V \times V$ such that for all $(X_t, Y_t) \in V \times V$ there exists a path $X_t = Z^0, \ldots, Z^r = Y_t$ between $X_t$ and $Y_t$ where $(Z^i, Z^{i+1}) \in E$ for $i \in [r - 1]$ and $\sum_i \delta(Z^i, Z^{i+1}) = \delta(X_t, Y_t)$. Suppose a coupling $(S, T) \to (S', T')$ of the Markov chain is defined on all pairs in $E$ such that there exists an $\alpha < 1$ such that $\mathbb{E}[\delta(S', T')] \leq \alpha\delta(S, T)$ for all $(S, T) \in E$, then we have $\tau(\varepsilon) \leq \frac{\log(D\varepsilon^{-1})}{(1-\alpha)}$.*

We now are ready to state our proof.

*Proof.* We define $\delta(X,Y) = \frac{1}{2}(|X \oplus Y| + ||X| - |Y||)$. It is clear that $\delta(X,Y) \geq 1$ for $X \neq Y$. Let $E = \{(X,Y) : \delta(X,Y) = 1\}$ be the set of adjacent states (neighbors), and it follows that $\delta(\cdot,\cdot)$ is a metric satisfying conditions in Lemma 7. Also we have $\delta(X,Y) \leq k$.

We consider constructing a path coupling between any two states $S$ and $T$ with $\delta(S,T) = 1$, $S'$ and $T'$ be the two states after transition. We sample $c_S, c_T \in \{0,1\}$, if $c_S$ is 0 then $S' = S$ and the same with $c_T$. $i_S, i_T \in V$ are drawn uniformly randomly. We consider two possible settings for $S$ and $T$:

1. If $S$ or $T$ is a subset of the other, we assume without of generality that $S = T \cup \{t\}$. In this setting we always let $i_S = i_T = i$. Then

    (a) If $i = t$, we let $c_S = 1 - c_T$;
        i. If $c_S = 1$ then $\delta(S',T') = 0$ with probability $p^-(S,t)$;
        ii. If $c_S = 0$ then $\delta(S',T') = 0$ with probability $p^+(T,t)$;
    (b) If $i \in T$, we set $c_S = c_T$;
        i. If $c_S = 1$ then $\delta(S',T') = 2$ with probability $(p^-(T,i) - p^-(S,i))_+$;
    (c) If $i \in V \backslash S$, we set $c_S = c_T$;
        i. If $c_S = 1$ and $|S| < k$ then $\delta(S',T') = 2$ with probability $(p^+(S,i) - p^+(T,i))_+$.

2. If $S$ and $T$ are of the same sizes, let $S = R \cup \{s\}$ and $T = R \cup \{t\}$. In this setting we always let $c_S = c_T = c$. We consider the case of $c = 1$:

    (a) If $i_S = s$, let $i_T = t$. Then $\delta(S',T') = 0$ with probability $\min\{p^-(S,s), p^-(T,t)\}$;
    (b) If $i_S = t$, let $i_T = s$. If $|S| < k$, Then $\delta(S',T') = 0$ with probability $\min\{p^+(S,t), p^+(T,s)\}$;
    (c) If $i_S \in R$, let $i_T = i_S$. Then $\delta(S',T') = 2$ with probability $|p^-(S,i_S) - p^-(T,i_T)|$;
    (d) If $i_S \in V \backslash (S \cup T)$, let $i_T = i_S$. If $|S| < k$, Then $\delta(S',T') = 2$ with probability $|p^+(S,i_S) - p^+(T,i_T)|$.

In all cases where we didn't specify $\delta(S',T')$, it will be $\delta(S',T') = 1$. In the first case of $S = T \cup \{t\}$ we have

$$\frac{\mathbb{E}[\delta(S',T')]}{\mathbb{E}[\delta(S,T)]} \leq \frac{1}{2N}((1 - p^-(S,t)) + (1 - p^+(T,t)) + (2|T| + \sum_{i \in T}(p^-(T,i) - p^-(S,i))_+) +$$

$$(2(N - |S|) + [\![|S| < k]\!] \sum_{i \in [N] \backslash S}(p^+(S,i) - p^+(T,i))_+))$$

$$= 1 - \frac{1}{2N}(1 - \sum_{i \in T}(p^-(T,i) - p^-(S,i))_+ - [\![|S| < k]\!] \sum_{i \in [N] \backslash S}(p^+(S,i) - p^+(T,i))_+) = 1 - \frac{\alpha_1}{2N},$$

while in the second case of $|S| = R \cup \{s\}$ and $T = R \cup \{t\}$ we have

$$\frac{\mathbb{E}[\delta(S',T')]}{\mathbb{E}[\delta(S,T)]} \leq \frac{1}{2N}((1 - \min\{p^-(S,s), p^-(T,t)\}) + (1 - [\![|S| < k]\!] \min\{p^+(S,t), p^+(T,s)\}) +$$

$$(2|R| + \sum_{i \in R}|p^-(S,i) - p^-(T,i)|) +$$

$$(2(N - |S| - 1) + [\![|S| < k]\!] \sum_{i \in [N] \backslash (S \cup T)}|p^+(S,i) - p^+(T,i)|))$$

$$= 1 - \frac{1}{2N}(\min\{p^-(S,s), p^-(T,t)\} - \sum_{i \in R}|p^-(S,i) - p^-(T,i)| +$$

$$[\![|S| < k]\!](\min\{p^+(S,t), p^+(T,s)\} - \sum_{i \in [N] \backslash (S \cup T)}|p^+(S,i) - p^+(T,i)|)) = 1 - \frac{\alpha_2}{2N}.$$

Let $\alpha = \max_{(S,T) \in E}\{\alpha_1, \alpha_2\}$. If $\alpha < 1$, with Lemma 7 we have

$$\tau(\varepsilon) \leq \frac{2N \log(k/\varepsilon)}{1 - \alpha}.$$

$\square$

# C Proof of Thm. 2

We recall key aspects of strongly Rayleigh distributions, on which our proof of fast mixing depends[5]. Let $\pi$ be a probability distribution on $\{0,1\}^N$, its *generating polynomial* is defined as

$$f_\pi(z) = \sum_{S \in [N]} \pi(S) z^S,$$

where $z = (z_1, \ldots, z_N)$ and $z^S = \prod_{i \in S} z_i$. One of useful properties of such polynomial is their stability. A polynomial $f \in \mathbb{C}[z_1, \ldots, z_N]$ is called *stable* if $f(z) \neq 0$ whenever $\mathcal{IM}(z_j) > 0$ for $j \in [N]$. A stable polynomial with all real coefficients is called *real stable*.

Strongly Rayleigh distribution is defined upon properties of its generating polynomial: A distribution $\pi$ is called *strongly Rayleigh* if its generating polynomial $f_\pi$ is (real) stable.

One of common manipulations on distributions over $\{0,1\}^N$ is symmetric homogenization, where one construct distributions on $\{0,1\}^{2N}$ such that their marginal distribution on $[N]$ is the same as the original ones on $\{0,1\}^N$.

**Definition 8** (Symmetric Homogenization). Given $\pi$ on $\{0,1\}^N$, define a new distribution $\pi_{sh}$ on $\{0,1\}^{2N}$ called the *symmetric homogenization* of $\pi$ by

$$\pi_{sh}(S) = \begin{cases} \pi(S \cap [N]) \binom{N}{S \cap [N]}^{-1} & \text{if } |S| = N; \\ 0 & \text{otherwise.} \end{cases}$$

The class of strongly Rayleigh distribution has been proved to be closed under symmetric homogenization:

**Theorem 9** (Closure under Symmetric Homogenization [4]). *If $\pi$ is strongly Rayleigh then so its symmetric homogenization $\pi_{sh}$.*

Strongly Rayleigh distribution includes many distributions such as DPP as special cases. Only recently, the Markov chain constructed for sampling from homogeneous strongly Rayleigh distribution has been proved to be rapidly mixing.

**Theorem 10** (Rapid Mixing for Homogeneous Strongly Rayleigh [3]). *For any strongly Rayleigh $k$-homogeneous probability distribution $\pi : \{0,1\}^N \to \mathbb{R}_+$, we have*

$$\tau_{X_0}(\varepsilon) \leq 2k(2N-k)(\log \pi(X_0)^{-1} + \log \varepsilon^{-1}).$$

*where $2N$ is the size of the ground set.*

Now we are ready to prove our statement in Thm. 2.

**Proof of Thm. 2** Given a strongly Rayleigh distribution $\pi_\mathcal{C}$, we construct its symmetric homogenization $\pi_{sh}$ as in Def. 8. By Thm. 9 we know that $\pi_{sh}$ is homogeneous strongly Rayleigh. Then it follows from Thm. 10 that the base exchange Markov chain has its mixing time bounded as

$$(\tau_{sh})_{Y_0}(\varepsilon) \leq 2N^2(\log(\pi_{sh}(Y_0))^{-1} + \log \varepsilon^{-1})$$
$$= 2N^2 \left( \log \binom{N}{|X_0|} + \log(\pi_\mathcal{C}(X_0))^{-1} + \log \varepsilon^{-1} \right),$$

where $Y_0 \subseteq [2N]$, $|Y_0| = N$ and $X_0 = Y_0 \cap V$.

We construct a base exchange Markov chain on $2N$ variables where we maintain a set $|R| = N$. In each iteration and with probability 0.5 we choose uniformly $s \in R$ and $t \in [2N]\backslash R$ and switch them with certain transition probabilities. Let $S = R \cap V$, $T = V\backslash R$, there are in total four possibilities for locations of $s$ and $t$:

1. With probability $\frac{|S|(N-|S|)}{2N^2}$, $s \in S$ and $t \in T$, and we switch assignment of $s$ and $t$ with probability $\min\{1, \frac{\pi_{sh}(R \cup \{t\}\backslash\{s\})}{\pi_{sh}(R)}\} = \min\{1, \frac{\pi_\mathcal{C}(S \cup \{t\}\backslash\{s\})}{\pi_\mathcal{C}(S)}\}$. This is equivalent to switching elements between $S$ and $T$;

2. With probability $\frac{|S|(N-|S|)}{2N^2}$, $s \notin S$ and $t \notin T$, and switch with probability $\min\{1, \frac{\pi_C(S \cup \{t\})}{\pi_C(S)} \times \frac{|S|+1}{N-|S|}\}$. This is equivalent to doing nothing to $S$;

3. With probability $\frac{|S|^2}{2N^2}$, $s \in S$ and $t \notin T$, and we switch with probability $\min\{1, \frac{\pi_C(S \setminus \{s\})}{\pi_C(S)} \times \frac{|S|}{N-|S|+1}\}$. This is equivalent to deleting elements from $S$;

4. With probability $\frac{(N-|S|)^2}{2N^2}$, $s \notin S$ and $t \in T$, and switch with probability $\min\{1, \frac{\pi_C(S \cup \{t\})}{\pi_C(S)} \times \frac{|S|+1}{N-|S|}\}$. This is equivalent to adding elements to $S$.

Constructing the chain in the same manner but only maintaining $S = R \cap [N]$ will result in Algo. 1, while the mixing time stays unchanged.

## D  Supplementary Experiments

### D.1  Varying $\delta$

We run 20-variable chain-structured Ising model on partition matroid base of rank 5 with varying $\delta$'s. The results are shown in Fig. 4 and Fig. 5. We observe that the approximate mixing time grows with $\delta$.

(a)                          (b)                          (c)

Figure 4: Convergence of marginal (`Marg`) and conditional (`Cond-1` and `Cond-2`, conditioned on 1 and 2 other variables) probabilities of a single variable in a 20-variable Ising model. We fix $\beta = 3$ and vary $\delta$ as (a) $\delta = 0.2$, (b) $\delta = 0.5$ and (c) $\delta = 0.8$. Full lines show the means and dotted lines the standard deviations of estimations.

(a)                          (b)                          (c)

Figure 5: PSRF of each set of chains in Fig. 4 with $\beta = 3$ and (a) $\delta = 0.2$; (b) $\delta = 0.5$ and (c) $\delta = 0.8$.

Figure 6: Comparisons of PSRF's for marginal estimations with different $\delta$'s. (a) PSRF's with different $\delta$'s and (b) the approximate mixing time estimated by thresholding PSRF at 1.05.

## D.2 Varying $\beta$

We run 20-variable chain-structured Ising model on partition matroid base of rank 5 with varying $\beta$'s. The results are shown in Fig. 7 and Fig. 8. We observe that the approximate mixing time grows with $\beta$.

Figure 7: Convergence of marginal (`Marg`) and conditional (`Cond-1` and `Cond-2`, conditioned on 1 and 2 other variables) probabilities of a single variable in a 20-variable Ising model. We fix $\delta = 1$ and vary $\beta$ as (a) $\beta = 0.5$; (b) $\beta = 2$ and (c) $\beta = 3$. Full lines show the means and dotted lines the standard deviations of estimations.

Figure 8: PSRF of each set of chains in Fig. 7 with $\delta = 1$ and (a) $\beta = 0.5$; (b) $\beta = 2$ and (c) $\beta = 3$.

## D.3 Varying Data Sizes

We run $(k\text{-})$DPP that is constrained to sample subsets from 1) partition matroid base and 2) uniform matroid with different data sizes $N$.

Figure 9: Comparisons of PSRF's for marginal estimations with different $\beta$'s. (a) PSRF's with different $\beta$'s and (b) the approximate mixing time estimated by thresholding of 1.05 on PSRF's.

### D.3.1  Partition Matroid Constraint

The estimations for marginal and conditional distributions are shown in Fig. 10 and corresponding PSRF's are shown in Fig. 11. We observe that the estimation becomes stable faster when $N$ is small.

Figure 10: Convergence of marginal (`Marg`) and conditional (`Cond-1` and `Cond-2`, conditioned on 1 and 2 other variables) probabilities of a single variable in a $k$-DPP on partition matroid base of rank 5, with (a) $N = 20$; (b) $N = 50$ and (c) $N = 100$. Full lines show the means and dotted lines the standard deviations of estimations.

Figure 11: PSRF of marginal (`Marg`) and conditional (`Cond-1` and `Cond-2`, conditioned on 5 and 10 other variables) probabilities of a single variable in a $k$-DPP on partition matroid base of rank 5, with (a) $N = 20$; (b) $N = 50$ and (c) $N = 100$.

### D.3.2 Uniform Matroid Constraint

The estimations for marginal and conditional distributions are shown in Fig. 12 and corresponding PSRF's are shown in Fig. 13. We observe the same thing as mentioned before, that the estimation becomes stable faster when $N$ is small.

(a)　　　　　　　　　　(b)　　　　　　　　　　(c)

Figure 12: Convergence of marginal (`Marg`) and conditional (`Cond-5` and `Cond-10`, conditioned on 5 and 10 other variables) probabilities of a single variable in a DPP on uniform matroid of rank 30, with (a) $N = 50$; (b) $N = 100$ and (c) $N = 200$. Full lines show the means and dotted lines the standard deviations of estimations.

(a)　　　　　　　　　　(b)　　　　　　　　　　(c)

Figure 13: PSRF of marginal (`Marg`) and conditional (`Cond-5` and `Cond-10`, conditioned on 5 and 10 other variables) probabilities of a single variable in a DPP on uniform matroid of rank 30, with (a) $N = 50$; (b) $N = 100$ and (c) $N = 200$.

## Footnotes

[5]Part of the material is drawn from [4], we include it for self-containness.