[Reviews · NeurIPS 2016]

Reviewer 1

Summary

The manuscript addresses the problem of sampling from distributions over subsets of a given set, e.g. determinantal point processes (DPPs). Ising models and other MRF models can also be cast in this form. The paper provides polynomial-time upper bounds on the mixing time for certain simple MCMC algorithms that sample from these models, under several different conditions on the model, and with the ability to handle constraints.

Qualitative Assessment

The theoretical results on MCMC mixing time presented in this paper are shown at a high level of mathematical abstraction. Nevertheless, at least some of these results have very clear practical importance. In particular, the special cases of these results that are applicable to DPPs (and k-DPPs) solve an important problem with practical significance for the probabilistic modeling of diverse sets. This is especially the case given that a previous paper (reference [23]) introduced practical MCMC algorithms for DPPs and aimed to give mixing time results, but had an erronous proof, making this a clear open problem of interest to the community. This special case alone establishes the value of this work. The experimental results are illustrative and are overall adequate, especially considering that this is a primarily theoretical paper. They could however be made stronger by including an application-oriented case study. While the manuscript is fairly dense, it is readable and well organized. Here are a couple of small typos that should be corrected on page 6: "the combination intuitively makes sens" "Algoritm 3" One apparent omission appears to be the proof that the proposed Markov chains in Algorithms 1, 2, and 3 are invariant to their target distributions. Presumably this could be verified straightforwardly using detailed balance? Nevertheless, for completeness, this should be shown in the supplementary, and mentioned in the main manuscript. Note that I have not checked the proofs.

Confidence in this Review

1-Less confident (might not have understood significant parts)


Reviewer 2

Summary

The paper is concerned with MC sampling of discrete probability distributions over sets with constraints. The authors present theoretical bounds for the mixing time of the Markov chain for three different algorithms that are applicable to three different classes of problems. The dependencies on variables in the bounds are illustrated empirically by experiments.

Qualitative Assessment

Technically the paper is very strong. The results presented by the authors are, to the best of my knowledge, novel and significant. However my main criticism of the paper is that the presentation is very esoteric. The is clear already in the introduction where the authors fail to explain some of the basic notation that is central to the remaining of the paper, see (1)-(3) below. This continues throughout the paper making it hard to read for non-experts in the field, see e.g. (8), (9) and (13) below. As a consequence of the esoteric presentation the authors fail to convince me of the broader relevance and the impact of the paper. Specific comments: (1) In line 17 the authors do not explain the notation [N] and the do not introduce the variable N. (2) In line 23 the authors do not explain the notation 2^N, which naturally has a different meaning. This means that it is actually unclear that C is a family of subsets of V=[N]. (3) In line 39-45 the authors assume that the reader is familiar with writing constraints as |S|=k and they do not introduce the variable k. (4) In line 38 it is not clear what the authors mean by ‘“old” and new Markov Chains”. Also the C should be lowercase. (5) In footnote 1 please elaborate on why the analysis in ref. [23] is not correct. (6) In Algorithm 1 I believe that “C \subseteq V” is a misprint. (7) In line 90 I believe “...and the second…” should be “...then the second…” (8) In line 100 I believe that f is never introduced in the paper. (9) In Theorem 3 please explain the \ni notation in the sum. (10) In the beginning of section 2 the ground set is denote V while in subsection 2.1 it is donated [N]. While they are equal by definition, is there a reason for the change in notation? (11) In Algorithm 2 the line “With probability 0.5, pick s \in [N] uniformly randomly” is very unclear. One has to read the text to understand that this line means that alternatively nothing is done. (12) In line 186 I believe “...set of size [2N]...” should be “...set of size 2N...”. (13) In lines 215 and 221-222 the authors never explain which marginal and conditional distributions they are considering. (14) In figure 1a, 1b, 1c and 3a the authors never explain the difference between the dotted and full lines.

Confidence in this Review

1-Less confident (might not have understood significant parts)


Reviewer 3

Summary

This paper analyzes the mixing time of MCMC sampling from discrete distributions under three novel settings 1.uniform+partition matroid constraints 2. uniform matroid constraints of certain rank, and 3. sampling from strongly Rayleigh distributions.

Qualitative Assessment

Pros: The paper is well-motivated and makes important theoretical contributions. Background literature is adequately reviewed and put into context while presenting the contributions. The proof techniques look sound and are provided in sufficient depth. Cons: 1. The theoretical results rely on strong assumptions on the structural parameters, which as the authors point out, are hard to quantify. I would have liked to see more discussion on what are the different settings under which these assumptions hold as well as those under which they do not hold true. 2. Experiments: a. Some experiments on DPPs with partition matroid would complement the theory. b. Figure 3: (i) Why are there negative values in Figure 3(a) ? also, have those chains really converged? (ii) It is unclear what are the tradeoffs that the Add-Delete and Mix chains are making. Why does a lazier strategy lead to faster mixing for certain class of DPPs and not others?

Confidence in this Review

1-Less confident (might not have understood significant parts)


Reviewer 4

Summary

The authors theoretically analyzed the mixing rate of sampling from discrete sampling distribution with constraint conditions, which is of general interest to machine learning community. The authors considered two cases where the $|S|$ is either equal to $k$ (matroid bases) or $ \leq k$ (uniform matroid). In each case the mixing time can be bounded. In Strongly Rayleigh distribution, the mixing time can be derived with an MC algorithm with 3 operations: exchange, add and delete.

Qualitative Assessment

I must confess that I am not an expert in Determinantal Point Processes (DPPs). I like that the authors start by first explaining the setup and connections with other methods. It shows the mixing times for both $|S|=k$ and $|S| \leq k$. Additionally, showing that in strongly Rayleigh distributions fast mixing can be obtained by mixing an add-delete chain with an exchange chain, though this is not very surprising. The authors used novel proof techniques. I have some confusion when I read the remarks for Theorem 3 and Corollary 6. In Theorem 3, the authors generalize the bound to any $\pi_c$, by including a factor $\exp(2\beta\zeta_F)$. In Corollary 6, they said that their bound is more general comparing to [18] because it's free of factor $\exp(2\beta\zeta_F)$, thus free for $F$. However in [3], the bound for SR distribution seems also free of $F$. I would be interested to see how the authors compare their bound of SR distribution with the bound in [3]. The paper is well structured. Minor comments: Page 4: Equation 2.4 does not align properly. Page 7: Figure 1: No instructions for the dashed lines. For each of the subfigures, the sampler does not seems like reaching convergence. The empirical illustration based on single observation seems not very convincing.

Confidence in this Review

1-Less confident (might not have understood significant parts)


Reviewer 5

Summary

The paper studies the mixing time of simple Markov chains on rather regular families of subsets of a ground set (e.g. the bases of a very special matroid), with somewhat arbitrary stationary distributions proportional to a given set function. The mixing time is shown to be about linear or quadratic in the size of the ground set, while exponential in a quantity that measures the nonlinearity of the set function (in log scale). Furthermore, it is shown that for so-called strongly Rayleigh distributions (over all subsets of the ground set), which include so-called Determinantal Point Processes as a special case, that exponential factor is avoided. Finally, experimental results demonstrate the effect of nonlinearity to mixing time.

Qualitative Assessment

The paper is technically of high quality. It cites lots of relevant literature, uses appropriate sophisticated techniques (multicommodity flow, path coupling), and is mathematically rigorous. The work can viewed as continuation of some recent studies, particularly the one by Anari et al. (COLT 2016). Unfortunately, the paper is not very explicit about what are the really new ideas in the present paper. However, this looks like a presentation issue; for example, concerning strongly Rayleigh distributions, the use of symmetric homogenization seems like a truly new and important idea. Fast-mixing Markov chains is generally a very important research topic, which so far has been investigated mostly from a theoretical point of view. The present work, while also rather theoretical, does include also a more practical perspective, which should be appreciated in the NIPS community. The presentation is generally fine, but it could be improved in several ways. Most importantly, the paper could be more explicit about what exactly was known about mixing times for the studied problems: does the present results improve upon the state of the art or are the existing results just incomparable? Minor suggestions/comments: - For simplicity, consider using \pi instead of exp(...) in Algorithms 1 and 2. - State the context in the theorem statements. Currently it is not clear what problem and algorithm are considered. - Discuss the relationship of Sections 2 and 3. Currently they look rather independent. - What are the dotted lines in Fig. 1?

Confidence in this Review

2-Confident (read it all; understood it all reasonably well)


Reviewer 6

Summary

This paper is concerned with sampling from a general class of discrete probability distributions subject to constraints and using Markov Chain methods. The paper makes three contributions relating to the analysis of mixing times under different constraints and the implicit constraint of the discrete measure being strongly Rayleigh, relating to fast mixing of determinantal point processes. The authors derive theoretical bounds and validate their results in a series of experiments on both simulated and real data.

Qualitative Assessment

This paper is rigorous and of high technical quality, making significant theoretical contributions and validating these using appropriate experiments. The theoretical results presented seem to be relevant to a larger literature on a number of probabilistic models and DPPs, providing theoretical results on mixing times when sampling from these distributions. The content of the paper may be less accessible to a wider audience, as it is mostly theoretical, though the results are novel and original. Overall, the paper is well-written and well-structured. The results are presented in a clear and coherent fashion.

Confidence in this Review

1-Less confident (might not have understood significant parts)


Reviewer 7

Summary

This paper presents and analyzes several Markov chains for sampling from various discrete probability distributions that allow for hard constraints. The authors analyze the mixing rate of the bases-exchange Gibbs sampler and the add-delete Gibbs sampler when the constraints are special types of matroid bases. The authors then go on to show that a modification of the bases-exchange Gibbs sampler can be shown to mix rapidly when the target distribution satisfies the condition of being Strongly Rayleigh, a condition which is satisfied by Determinantal Point Processes. Finally, experimental results are given which empirically demonstrate that the theoretical bounds presented depend on the correct quantities.

Qualitative Assessment

This paper is technically sound and addresses interesting questions relating to sampling and mixing rates. In fact, this paper generalizes two previous results: (a) Section 2 can be considered an extension of Gotovos et al. (NIPS 2015). Unlike that paper, however, this paper allows for hard constraints. It is quite remarkable that even with these hard constraints, the authors are able to recover the same mixing rates as Gotovos et al. (b) Section 3 can be thought of as an extension of Anari et al. (COLT 2016), but the difference is that Anari et al. could only handle homogeneous Strongly Rayleigh distributions, while this paper can handle any Strongly Rayleigh distribution. The proof, in which a Strongly Rayleigh distribution is viewed as the marginal distribution of a homogeneous Strongly Rayleigh distribution over a larger set of elements, is quite clever. As best as I can tell, all the statements given in the paper and all the final bounds in the proofs presented in the supplementary material are essentially correct, modulo some constant factors. HOWEVER, there are some grave typos and undefined notation which make the intermediate steps in the proofs unintelligible. Detailed comments: 1. In the proofs for Theorem 2, it appears that a factor of 2 is missing due to the fact that they are using the lazy version of the Markov chain. Thus, both of the mixing rate bounds should be increased by a factor of 2. 2. In the equations following lines 340 and 370, the factor of Z should appear in the denominator and not the numerator. Fortunately, this appears to be a typo and does not propagate through the rest of the proofs. 3. In equation (2.3) and all of appendix A, every occurrence of exp(\beta(F(\sigma_S(X,Y)) + F(\sigma_T(X,Y)))) should be replaced with exp(\beta F(\sigma_S(X,Y))) + exp(\beta F(\sigma_T(X,Y))). Similarly, every occurrence of exp(\beta(F(U_S) + F(U_T))) should be replaced with exp(\beta F(U_S)) + exp(\beta F(U_T)). This too appears to be a typo and does not propagate through the rest of the proofs. 4. In appendix A.2 and A.3, the quantity Z_F is never defined, although it seems to be taken as \sum_{U_S} exp(\beta F(U_S)). 5. In Theorem 4, it ought to be made explicit that \alpha_1 and \alpha_2 are functions of edges (S,T). 6. In Theorem 4 and the proof in appendix B, the notation | |_+ and ( )_+ ought to be defined. 7. In the proof of Theorem 5, it should to be made clear that Theorem 10 is being instantiated with N := 2N and k := N. 8. In the figures of section 4 and appendix D, the meaning of the dashed lines ought to be made clear.

Confidence in this Review

3-Expert (read the paper in detail, know the area, quite certain of my opinion)